# Characterization and Preliminary Safety Evaluation of *Akkermansia muciniphila* PROBIO

**DOI:** 10.3390/foods13030442

**Published:** 2024-01-30

**Authors:** Xin Ma, Meng Tian, Xueping Yu, Ming Liu, Bin Li, Dayong Ren, Wei Wang

**Affiliations:** 1State Key Laboratory of Bioreactor Engineering, East China University of Science and Technology, Shanghai 200237, China; pkartest@yeah.net (X.M.); yuxueping@thankcome.com (X.Y.); 2College of Food Science and Engineering, Jilin Agricultural University, Changchun 130118, China; jodie0730@163.com; 3China National Research Institute of Food and Fermentation Industries, Beijing 100015, China; lm_bob@163.com (M.L.); libin_2957@163.com (B.L.)

**Keywords:** *Akkermansia muciniphila*, safety, toxicity, genome, antibiotic

## Abstract

In addition to providing certain health advantages to the host, a bacterial strain must possess a clearly defined safety profile to be regarded as a probiotic. In this study, we present a thorough and methodical assessment of the safety of a novel strain of bacteria, *Akkermansia muciniphila* PROBIO, which was isolated from human feces. Firstly, we examined the strain’s overall features, such as its gastrointestinal tolerance and its physiological and biochemical traits. Next, we verified its genotoxic properties through bacterial reverse mutation and in vitro mammalian cell micronucleus assays. The drug sensitivity of *A. muciniphila* PROBIO was subsequently examined through an analysis of its antibiotic resistance genes. Additionally, the toxicological impact was verified through acute and sub-chronic toxicity studies. A genome-based safety assessment was conducted to gain further insights into gene function, including potential virulence factors and pathogenic properties. Finally, we assessed whether moxifloxacin resistance in *A. muciniphila* PROBIO is transferred using in vitro conjugation experiments. *A. muciniphila* PROBIO exhibited superior gastrointestinal tolerance, with no observed hematological or histopathological abnormalities. Moreover, the outcomes pertaining to mutagenic, clastogenic, or toxic impacts were found to be negative, even at exceedingly high dosages. Moreover, no adverse effects associated with the test substance were observed during the examination of acute and sub-chronic toxicity. Consequently, it was plausible to estimate the no-observed-adverse-effect level (NOAEL) to be 6.4 × 10^11^ viable bacteria for an average individual weighing 70 kg. Additionally, only three potential drug resistance genes and one virulence factor gene were annotated. *A. muciniphila* PROBIO is naturally resistant to moxifloxacin, and resistance does not transfer. Collectively, the data presented herein substantiate the presumed safety of *A. muciniphila* PROBIO for its application in food.

## 1. Introduction

*Akkermansia muciniphila*, a member of the phylum *Verrucomicrobia*, is an oval-shaped, Gram-negative, and strictly anaerobic bacterium that resides as a commensal in the human intestinal tract. This type of bacterium is immobile and lacks the ability to form spores. *A. muciniphila* was first isolated from the human intestine in 2004 [1] and has been found in the small and large intestines, with a particularly high abundance in the cecum, where most of the mucin is produced [2]. The proportion of *A. muciniphila* in the intestinal microbiota of humans and other mammal species ranges from 3% to 5% [3]. Additionally, *A. muciniphila* has been detected in the breast milk and breast tissue of lactating women, utilizing breast milk oligosaccharides as a source of energy, carbon, and nitrogen [4]. This bacterium is vertically transmitted from breast milk to infants and plays a role in the development of the intestinal microbiota during early infancy (around 1 month of age), with a significant increase in abundance during adulthood [3]. A recent comprehensive analysis of a large global dataset revealed that *A. muciniphila* was present in 77.73% of the global evaluated cohorts and in 81.81% of the western population cohorts [5].

Although *A. muciniphila* has only been isolated for a short time, it has been extensively researched for its impact on human health. *A. muciniphila* produces a group of enzymes that specialize in the degradation of mucins on the surface of the human gastrointestinal tract [6]. This degradation of human mucus proteins by *A. muciniphila* serves as a competitive exclusion mechanism against pathogenic mucus degradants and may have gatekeeper and signaling functions [7,8]. The decomposition of mucus leads to the production of beneficial metabolites, such as acetic acid, propionic acid, and oligosaccharides, contributing to improved metabolism and reduced inflammation [9,10]. *A. muciniphila* also influences mucus production by regulating the differentiation of intestinal epithelial cells to improve the integrity of the intestinal barrier and mucosal homeostasis [11]. As *A. muciniphila* has been found to be abundant in the intestines of healthy individuals and reduced in patients with inflammatory bowel disease, obesity, and other disorders, it has been proposed as a potential biomarker for intestinal health [9,12]. A large number of studies have shown that changes in the abundance of *A. muciniphila* are related to metabolic disorders [13,14], immune diseases [15,16], and cancer [17]. Notably, oral administration of live or pasteurized *A. muciniphila* can regulate the metabolism of the host, resulting in a decrease in serum triglyceride levels and an improvement in insulin sensitivity [18]. This intervention was shown to offer a certain degree of protection against diet-induced obesity, diabetes, and intestinal barrier dysfunction in mice. Furthermore, studies have demonstrated an elevation in mucins among individuals who showed positive reactions to immune checkpoint inhibitors [19]. Additionally, the presence of *A. muciniphila* has been found to augment the anti-tumor efficacy of Programmed Cell Death Protein 1 (PD-1) blockers [20]. Given the diverse advantageous biological properties exhibited by *A. muciniphila* in relation to human health, it is widely acknowledged as a promising candidate for novel pharmaceutical development or as a next-generation probiotic [21].

Despite the availability of data supporting the positive effects of *A. muciniphila*, the lack of thorough safety research is mostly to blame for the strain’s absence from the list of strains that have been certified by governmental agencies in many nations across the world. Several preclinical and clinical studies have indicated that *A. muciniphila* is predominantly eliminated through fecal excretion following oral administration, and supplementation of *A. muciniphila* would not cause substantial changes in the overall composition of the gut microbiome [13,22]. In 2021, the European Food Safety Authority (EFSA) issued a scientific opinion on the safety of pasteurized *A. muciniphila* as a novel food product [23]. This viewpoint led to the conclusion that the daily consumption of 3.4 × 10^10^ cells of pasteurized *A. muciniphila* was safe for the target population. Nevertheless, these findings alone do not provide adequate evidence to support the safety of *A. muciniphila* for use as a probiotic. In accordance with the guidelines set forth by the Food and Agriculture Organization (FAO) and the World Health Organization (WHO), probiotic strains must meet specific safety criteria. While probiotic strains derived from species typically present in fermented foods are generally deemed safe, strains originating from nontraditional species may elicit heightened concerns regarding potential adverse effects [24]. Therefore, further investigation is needed to ensure their safety. In our previous study, we successfully isolated a novel strain of *A. muciniphila* PROBIO (AKK PROBIO, AKK. pro) from human fecal samples. Our preliminary unpublished data showed that this strain exhibited promising health advantages, including the ability to reduce colitis symptoms, boost PD-1 antibodies, and produce short-chain fatty acids. The current study aimed to evaluate the initial safety of the novel *A. muciniphila* PROBIO at the strain level. The assessment encompassed an examination of various aspects of the strain, including the physicochemical characteristics of the cell surface, biofilm formation, resistance to antibiotics, tolerance to simulated gastrointestinal fluids, mutagenicity, acute toxicity, and sub-chronic toxicity.

## 2. Materials and Methods

### 2.1. Bacterial Strains and Culture Conditions

*A. muciniphila* PROBIO was isolated from the feces of a healthy adult and identified using 16S rRNA gene sequence analysis. This strain is currently stored in the China General Microbiological Culture Collection Center (Beijing, China) under the preservation number CGMCC No. 20955. AKK PROBIO was cultured in Brain Heart Infusion (BHI) broth medium (BD, Sparks, MD, USA) for 48 to 72 h at 37 °C under anaerobic conditions in a Basic1 anaerobic workstation (ChongQing Jiangxue Science and Technology, Chongqing, China) for oxygen deprivation (80% N_2_, 15% CO_2_, 5% H_2_). *Enterococcus faecalis* ATCC 29212 was used as a recipient, which was grown in Mueller-Hinton (MH) broth medium (BD, Sparks, MD, USA) at 37 °C for 24 h under anaerobic conditions.

### 2.2. Tolerance to Artificial Gastric and Intestinal Fluids

Gastrointestinal tolerance analysis refers to the method of Ulsemer [25]. The gastric juice (5 M HCl, 2 g/L NaCl, 3.2 g/L pepsin, pH 2.0) and intestinal juice (6.8 g/L KH_2_PO_4_, 10 g/L pancreatin, 0.1 M NaOH, pH 8.0) were prepared, followed by filtration (0.22 μm) and sterilization. The AKK PROBIO was activated twice in a 5 mL BHI liquid culture medium, then centrifuged at 3337× *g* for 10 min. The resulting bacterial precipitation was washed twice with sterile physiological saline and resuspended in 5 mL of physiological saline. The bacterial liquid was combined with sterile artificial gastric juice or intestinal juice (1:1, *v*/*v*). The mixture was thoroughly shaken and subsequently placed in a constant-temperature incubator (37 °C). The viable bacteria count was determined at 0 h and 4 h. The survival rate of AKK PROBIO in artificial gastric juice or intestinal juice was calculated using Formula (1).
(1)$$Survival\;rate=\frac{4\;h\;viable\;count\;(\mathrm{CFU}/\mathrm{mL})}{0\;h\;viable\;count\;(\mathrm{CFU}/\mathrm{mL})}$$

### 2.3. Evaluation of Hydrophobicity, Auto-Aggregation, and Biofilm Formation

Bacterial adhesion to hydrocarbons (BATH) was used to evaluate the cell surface hydrophobicity of AKK PROBIO [26]. Specifically, *A. muciniphila* PROBIO was cultured in BHI medium and incubated overnight at 37 °C. Cell precipitates were obtained through centrifugation (4 °C; 7104× *g*; 10 min), washed twice, and resuspended in phosphate buffer (PBS, pH 7). The absorbance (OD_600_ nm) was adjusted to 0.5 in order to standardize the bacterial count. The biphasic mixture was thoroughly homogenized by adding an equivalent amount of xylene and vortexing it for 5 min. The aqueous phase was then extracted, and its absorbance at 600 nm was measured after an incubation time at room temperature at intervals of 15, 30, and 60 min. The hydrophobicity of AKK PROBIO was calculated using Formula (2).
(2)$$Hydrophobicity\;\%=\frac{p0-p1}{P1}\times 100$$
where *p*0 and *p*1 are the absorbance values before and after extraction with hydrocarbons, respectively.

For auto-aggregation [26], the cell suspension of AKK PROBIO was prepared as described above and incubated at 37 °C. The absorbance was adjusted (OD_600_ nm) to 0.5. The bacterial suspension was cultivated at 37 °C, and the absorbance at 600 nm was read after intervals of 4, 8, 12, 16, 20, and 24 h. The auto-aggregation of AKK PROBIO was calculated using the following Formula (3):(3)$$Auto-aggregation\;\%=1-(\frac{p1}{P0})\times 100$$
where *p*0 is the absorbance at time 0, and *p*1 is the absorbance detected after 4, 8, 12, 16, 20, and 24 h.

For biofilm formation [27], AKK PROBIO was cultured overnight in tryptone soy broth (TSB) (BD, Sparks, MD, USA) at 37 °C. The bacterial cells were then harvested through centrifugation at 7104× *g* for 10 min at 4 °C, followed by two washes with PBS. The cells were subsequently resuspended in different media, including TSB medium without glucose and TSB medium with varying concentrations of glucose (0.25%, 1%, and 2.5%, *w*/*v*). Each bacterial suspension (200 μL) was aliquoted into a 96-well plate, with the uninoculated culture medium serving as the negative control. The plate was incubated at 37 °C for 24 h. The culture medium was aspirated from each well and subsequently washed three times with sterile physiological saline in order to eliminate any unattached cells. The attached cells were then fixed with methanol (200 μL per well) and allowed to dry for 15 min. Subsequently, the cells were subjected to staining with 2% crystal violet (200 μL per well) for a duration of 5 min. Following this, the wells were washed three times with sterile physiological saline to remove any excess dye. Once the plate had air dried, the attached cells were resuspended in 160 μL of glacial acetic acid (33%, *v*/*v*), and their absorbance at 600 nm was measured. The critical value, denoted as OD_0_, was established as the mean optical density (OD) value of the negative control. Based on the OD measurement, the strain was categorized into four groups: non-biofilm producers (OD ≤ OD_0_), weak biofilm producers (OD_0_ < OD ≤ 2 × OD_0_), medium biofilm producers (2 × OD_0_ < OD ≤ 4 × OD_0_), or strong biofilm producers (4 × OD_0_ < OD) [27].

### 2.4. Bacterial Reverse Mutation Test (Ames Test)

The bacterial reverse mutation test was performed according to the methods of plate incorporation and preincubation [28,29]. The bacterial strains employed in this study included *Salmonella Typhimurium* TA97a, TA98, TA100, and TA1535, and *Escherichia coli* WP2 uvrA. Distilled water was utilized as the carrier to create a bacterial suspension (100 mg/mL). *A. muciniphila* PROBIO, as the test item, had the potential to influence the test outcomes in various ways, thereby compromising the validity of the study. Consequently, the bacteria present in the test sample were eliminated through mechanical methods. During the preparation process, the soluble components of the test items are dissolved in the carrier and subsequently subjected to centrifugation at 4 °C and 8000 rpm for 10 min to eliminate any remaining bacteria. Following centrifugation, the supernatant was aseptically filtered using a 0.22 μm filter. Subsequently, a lower concentration preparation was prepared and utilized for the treatment of the test bacterial strain. 

The initial mutation study (plate incorporation method) and the confirmed mutation study (pre-culture procedure) were conducted based on the findings from preliminary experiments. Confirmatory mutation studies were conducted at concentrations of 15.81, 50, 158.1, 500, 1581, and 5000 μg/plate, adhering to the highest concentration advised by the prevailing regulatory guidelines. The positive controls encompassed Dixon, sodium azide, methyl methane sulfonate, and 2-aminofluorene, which were dissolved in either distilled water or dimethyl sulfoxide. The survival rate of the test bacterial cells was checked by the plate experiment in each test. If there was an increase in the number of responses dependent on the dosage, it could be inferred that the test item exhibited mutagenic properties. This inference holds true when at least one dose group demonstrates reproducible positive reactions that are biologically related, regardless of the presence or absence of metabolic activation. Additionally, if the number of reversals observed in *S. typhimurium* TA98, TA100, and *E. coli* WP2 uvrA strains was more than twice the number observed in the negative control, it was considered to be biologically significant. In the case of the *S. typhimurium* TA97a and TA1535 strains, if the number of reversals was more than three times the number observed in the negative control, it was also considered to be biologically significant.

### 2.5. In Vitro Mammalian Cell Micronucleus Test

The genotoxic potential of AKK PROBIO in inducing micronucleus formation in mouse erythrocytes was assessed using the cytokinesis-block method [30]. The Kunming (Aneta, #192) mice used in this study were obtained from the animal experimental center of Jilin Agricultural University and were approximately 7 weeks old. A 5-day adaptation period was provided for the animals, and only those in acceptable health were included in the study. The animals had access to distilled water. They were housed either in groups of five animals per cage or two animals per cage and were exposed to a 12-h light cycle.

In each experimental group, a total of five male mice were administered bacterial suspension at doses of 222.2, 666.7, and 2000 mg/kg body weight. The negative control group received distilled water, while the positive control group was administered cyclophosphamide at a dose of 60 mg/kg body weight. Following euthanasia, bone marrow samples were promptly collected from the femur of each mouse and prepared for glass slides, which were subsequently stained and evaluated for scoring purposes.

### 2.6. Antibiotic Susceptibility Test

The agar dilution method recommended by the Clinical and Laboratory Standards Institute (CLSI) was employed to assess the susceptibility of AKK PROBIO to clindamycin, meropenem, tetracycline, moxifloxacin, chloramphenicol, and ceftriaxone [31]. Similarly, the microbroth dilution method recommended by CLSI was utilized to determine the quality control strain, *Bacteroides fragilis* ATCC 25285 or *Staphylococcus aureus* ATCC 29213. Additionally, two databases, the CARD (The Comprehensive Antibiotic Resistance Database, https://card.mcmaster.ca/ accessed on 6 April 2023) and ResFinder (https://cge.cbs.dtu.dk/services/ResFinder/, accessed on 6 April 2023), were employed to predict potential drug resistance-related genes in *A. muciniphila* PROBIO.

### 2.7. Oral Acute and 90-Day Sub-Chronic Toxicity Studies

In order to evaluate the acute toxicity of AKK PROBIO, KM mice under specific pathogen-free (SPF) conditions were randomly divided into four groups (*n* = 8). Each group was given a suspension of AKK PROBIO (1 × 10^9^, 5 × 10^10^, and 5 × 10^11^ Colony-Forming Units (CFU)/day, respectively) or saline (0.5 mL/day) by oral gavage for 1 day, and their general health and weight were observed every day for 7–10 days. At the end of the experimental period, blood samples were collected for hematology and serum biochemical analysis. All animals in each group were examined for histopathology of major organs and tissues. The protocols were approved by the ethics committee of the Laboratory Animal Center of Jilin Agricultural University (permit number: 20220711002). 

To assess the potential systemic toxicity of AKK PROBIO, a sub-chronic oral toxicity study was conducted in vivo following the guidelines provided by the OECD test guidelines [32]. A total of 64 KM mice (32 males and 32 females) were obtained from the animal experimental center of Jilin Agricultural University. The administration of AKK PROBIO began when the animals were approximately 6–7 weeks old. The experimental environment was maintained at a room temperature of 22 ± 3 °C and a relative humidity of 55% ± 15%. The animals were divided into four groups in a random manner: group 1 only received saline; group 2 received a bacterial suspension at a dosage of 9.2 × 10^8^ CFU/kg body weight per day; group 3 received a bacterial suspension at a dosage of 27.6 × 10^8^ CFU/kg body weight per day; and group 4 served as a control. The animals were administered the suspension orally using a gavage technique for a period of 90 days. Daily records of clinical signs were maintained, and comprehensive clinical observation was conducted once a week. Additionally, the animals’ weight, food consumption, and water consumption were recorded on a weekly basis.

### 2.8. Genome Sequencing, Assembly, and Analysis

Genomic DNA extraction, determination of the quality and quantity of extracted DNA, DNA fragmentation, and sequencing with the PacBio Sequel II platform (Frasergen Bioinformatics Co., Ltd., Wuhan, China) were carried out. The raw sequencing reads (also called polymerase reads) generated from the PacBio platforms were processed using SMRTlink v8.0. Error correction of the reads was performed as part of the Canu (version 2.0) assembly process. The genome was assembled by Canu v2.0. Canu detects and annotates that the final assembly is circular when the best overlap graph (BOG) is circular. As an alternative method, we confirmed genome circularization by identifying repeated sequences at the two ends using BLASTN (version 2.7.1). The NCBI Prokaryotic Genome Automatic Annotation Pipeline (PGAP) was applied to complete genome annotation. The virulence factors and pathogenicity of AKK PROBIO were predicted using the VFD database (virulence factors of pathological bacteria, http://www.mgc.ac.cn/VFs/main.htm, accessed on 6 April 2023), VirulenceFinder database (https://cge.cbs.dtu.dk/services/VirulenceFinder, accessed on 6 April 2023), and PathogenFinder database (https://cge.cbs.dtu.dk/services/PathogenFinder, accessed on 6 April 2023). 

### 2.9. In Vitro Conjugation Experiments

The in vitro conjugation was carried out following the methods of Ashwini [33] to determine the transferability of the *adeF* gene of moxifloxacin resistance detected in AKK PROBIO (donor). *E. faecalis* ATCC 29212 (recipient) was used as a recipient to perform the conjugation in vitro. Briefly, from the overnight-grown cultures of the donor and the recipient, the cultures (1 mL) were centrifuged at 5000× *g* for 5 min to collect the bacterial cells. They were washed three times with PBS and resuspended to prepare a bacterial suspension of 0.5 on the McFarland scale. AKK PROBIO was thoroughly mixed with ATCC 29212 in volume ratios of 1:1 and 10:1, respectively. 

We inoculated the mixtures (donor/recipient ratios of 1:1 and 10:1) into MH liquid culture medium at a ratio of 2% (*v*/*v*), which were then incubated at 37 °C for 24 h. 

We took 0.1 mL of the mixtures (donor/recipient ratios of 1:1 and 10:1), spread them on to MH agar medium, and incubated them at 37 °C for 24 h. We then collected all colonies and resuspended them with PBS for a 10-fold gradient dilution.

The mixtures (donor/recipient ratios of 1:1 and 10:1) were filtrated through a sterile filter (0.22 μm). After the collection of donor and recipient cells on the filter, PBS (5 mL) was passed through the filter. We placed the filter membrane containing bacteria on MH agar medium and cultured them at 37 °C until colonies grew on the surface of the membrane. The bacterial colonies on the filter membrane were washed with PBS, and a dilution gradient was performed.

The transferability was detected on selective MH agar plates containing different concentrations of moxifloxacin. The bacterial solution of *E. faecalis* ATCC 29212 without AKK PROBIO was used as a negative control. Three parallel tests were run in each case.

### 2.10. Statistical Analysis

All experiments were performed at least in triplicate using independent assays, and values were expressed as the mean ± standard error. Statistical analysis was performed through the analysis of variance (ANOVA), followed by Tukey’s multiple comparison. A *p*-value of <0.05 was considered statistically significant. The software Graph Pad Prism (version 8.0) was used for the analysis.

## 3. Results

### 3.1. Gastrointestinal Tract (GI) Tolerance and Surface Properties

We determined the survival of *A. muciniphila* PROBIO after exposure to artificial gastric and intestinal fluids in vitro. After 240 min of exposure, the survival rates of AKK PROBIO in gastric and intestinal fluids were 84% (Figure 1a) and 82% (Figure 1b), respectively, suggesting that AKK PROBIO has good GI tolerance. The maximum values of hydrophobicity and auto-aggregation capacity of AKK PROBIO were 31% (60 min, Figure 1c) and 52% (24 h, Figure 1d), respectively. The ability of AKK PROBIO to form biofilms was weak regardless of the amount of glucose used. The results of these in vitro tests suggested that AKK PROBIO has a low ability to colonize the gut.

### 3.2. Bacterial Reverse Mutation Test

As shown in Table 1, within the confirmatory mutation study employing the plate incorporation technique, *E. coli* WP2 uvrA exhibited the highest rate of reversion when subjected to metabolic activation S9. Nevertheless, no discernible dose–response relationship was observed, and the quantities of revertant colonies fell within the range of historical controls. The mean values of revertant colony numbers observed in the untreated, negative, and positive control plates were found to fall within the historical control range for all strains. The reference mutagens exhibited a noticeable increase in the number of induced revertant colonies in each strain, regardless of the presence or absence of metabolic activation. The results indicated that, under the specific experimental conditions employed, the test item did not induce gene mutations through alterations in base pairs or frameshifts within the genomes of the strains utilized. AKK PROBIO showed no mutagenic activity in the examined bacterial strains under the present study conditions.

### 3.3. Micronucleus Analysis

The findings of the in vivo mammalian erythrocyte micronucleus study indicated that exposure to AKK PROBIO in mice did not result in any treatment-related impacts on body weight, mortality, systemic toxicity, or observable symptoms. The examination of bone marrow slides in all experimental groups involved the quantification of polychromatic erythrocytes (PCEs) and mature red blood cells in order to evaluate the incidence of micronuclei. The findings are presented in Table 2, indicating that there was no significant disparity in the occurrence of micronuclei in PCEs between the negative control and test groups (*p* > 0.05). However, the positive control treatment exhibited a substantial and statistically significant elevation in the frequency of micronuclei in PCEs (*p* < 0.001). The occurrence of micro-nucleated PCEs among all groups showed no significant differences. The results suggested that AKK PROBIO did not display genotoxic activity.

### 3.4. Antibiotic Resistance

The antibiotic resistance analysis results are presented in Table 3. AKK PROBIO exhibited susceptibility to clindamycin, meropenem, tetracycline, chloramphenicol, and ceftriaxone. However, it was resistant to moxifloxacin. The predicted results of drug resistance-related genes are shown in Appendix A. In the genome of AKK PROBIO, two genes associated with antibiotic resistance were identified, specifically pertaining to quinolones and aminoglycosides. One of these genes, *adeF*, is part of the gene cluster *adeFGH*. When the genes *adeF*, *adeG*, and *adeH* within the gene cluster are simultaneously expressed, they facilitate the encoding and production of the multidrug-resistant efflux pump *AdeFGH*, thereby conferring resistance to fluoroquinolones and other pharmaceutical agents upon the strain. In the ResFinder database, only one putative antibiotic resistance-related gene, *aadA15*, was identified, which was linked to aminoglycoside antibiotic resistance (Appendix A).

### 3.5. Acute and Sub-Chronic Toxicities

We found that none of the indices of the mice fed with AKK PROBIO at any given concentration were significantly different from the healthy control group in either the acute toxicity test or the sub-chronic toxicity test. Specifically, no treatment-related deaths or behavioral abnormalities were observed in the mice during the experiments. Figure 2a–d displays body weights and food consumption during the entire administration period. There was no significant difference in weight gain or food intake in all groups (*p* > 0.05) (Figure 2a–d). There were no significant changes in the blood glucose level (Figure 2e,f), glucose tolerance (Figure 2g), or insulin tolerance (Figure 2h) in all groups after 90 days of oral administration of AKK PROBIO (*p* > 0.05). There were no significant differences observed in red blood cells (RBCs), hemoglobin (HGB), platelets (PLTs), or other routine blood parameters among the groups (*p* > 0.05) (Table 4). Similarly, no significant differences were found in the biochemical markers associated with glucose and lipid metabolism in either the serum or liver across the groups (*p* > 0.05) (Table 5). The weights of organs in all groups showed no significant differences (*p* > 0.05) (Table 6). Furthermore, the structural integrity of the liver and renal tissues was unaffected by AKK PROBIO (Figure 3). All of these findings indicated that AKK PROBIO is safe for long-term usage at high doses (9.2 × 10^8^–2.76 × 10^9^ CFU/kg bw/day).

### 3.6. Genetic Characteristics

The complete genome sequences of *Akkermansia muciniphila* PROBIO are available at NCBI SRA accession number PRJNA1065580. The genome of AKK PROBIO consists of a single chromosome measuring 2,863,710 base pairs, with a GC content of approximately 55.32% (Appendix A). Utilizing the integrated data from genome sequencing, gene prediction, and non-coding RNA prediction, a gene circle diagram was constructed (Figure 4). To gain further insights into the functional annotation of genes, searches were conducted in the COG, KEGG, and GO databases. Out of the total predicted genes, 59.10% were successfully assigned to COG categories (Figure 5a). Among these assignments, 210 genes were associated with general function prediction, 195 genes were linked to cell wall, cell membrane, and envelope synthesis, 182 genes were involved in translation, ribosome structure, and biosynthesis, and 168 genes were associated with amino acid transport and metabolism. Additionally, 38.91% of the genes could be assigned to KEGG annotation (Figure 5b), with a total of 666 genes found to be related to metabolism. Specifically, 126 genes were involved in carbohydrate metabolism, 125 genes were associated with amino acid metabolism, and 102 genes were linked to functional factors and vitamin metabolism. The GO analysis revealed that there were 1322 genes associated with molecular function, 1332 genes associated with cellular components, and 1439 genes involved in biological processes (Figure 5c). Only one virulence factor gene, *tufA*, was identified in the VFD database (Appendix A). According to the PathogenFinder database, AKK PROBIO was predicted to be a non-human pathogen (Appendix A). These results indicated that AKK PROBIO was non-toxigenic at the genetic level. 

### 3.7. In Vitro Conjugal Transfer of Moxifloxacin Resistance

The results of moxifloxacin resistance gene transfer under different conditions for two donor/receptor ratios (1:1 and 10:1) are shown in Table 7. The MIC value of receptor strain *E. faecalis* 29212 for moxifloxacin did not change under different conditions of conjugation, indicating that the recipient strain did not obtain the moxifloxacin resistance gene from *A. muciniphila* PROBIO. Therefore, these results suggested that the moxifloxacin resistance gene from the donor strain AKK PROBIO did not transfer.

## 4. Discussion

Clear safety is a prerequisite for bacterial strains to be classified as probiotic strains. Consequently, probiotics such as *Bifidobacterium* and *Lactobacillus*, which are widely acknowledged as safe for human consumption, have garnered significant interest in both probiotic research and industry [34]. *A. muciniphila* is abundantly found in the human intestine and breast milk. Numerous studies have demonstrated a correlation between the absence of or reduction in *A. muciniphila* and various diseases, including obesity [35,36], diabetes [37], hepatic steatosis [38], inflammation [15], and the efficacy of immunotherapy [39]. Following the administration of *A. muciniphila*, a partial alleviation of symptoms associated with these diseases has been observed, thereby suggesting its potential as a next-generation probiotic. However, the safety assessment of *A. muciniphila* remained limited; only *A. muciniphila* ATCC BAA-835, DSM 22959, and AM02/06 have been evaluated and confirmed in animals [4,22,26]. A randomized controlled trial that applied up to 10^10^ CFU/day of live or pasteurized *A. muciniphila* ATCC BAA-835 found no adverse events or significant changes in clinical safety parameters for individuals with excess body weight [22]. However, the EFSA did not support this claim due to the limited subjects, disease scope, and short test period of the research [23]. The limited safety assessment of *A. muciniphila* in comparison to conventional probiotics hampers its utilization in clinical investigations and its potential for probiotic development. Therefore, it is imperative to thoroughly consider and evaluate the safety of other *A. muciniphila* strains.

A new *A. muciniphila* strain (*A. muciniphila* PROBIO) with potential probiotic properties was isolated from human feces. Given that *A. muciniphila* is not categorized as an opportunistic pathogen, this newly isolated strain has promise in terms of safety. Furthermore, the species demonstrates the ability to secrete a diverse range of enzymes, thereby augmenting its purported capacity for promoting health [40]. With the aim of carrying out a preliminary safety evaluation of a new strain that has no history of use in the food industry, we systematically evaluated and considered the safety of AKK PROBIO at the strain level, including the genetic information and physiological and biochemical characteristics of the strain, and also carried out an extensive toxicological evaluation.

The potential pathogenicity of *A. muciniphila* is related to adherence, which is related to the initial pathogenic behavior and the degradation of mucosal layers [41]. In this study, we evaluated the adhesion potential of AKK PROBIO to the intestinal mucosa. AKK PROBIO showed weak adhesion compared to other conventional probiotics, which indicates that it can colonize the intestine for only a short period of time. Notably, certain health effects of probiotics are thought to be influenced by their ability to adhere to the gut epithelium. However, as long as regular bacterial consumption takes place, probiotic colonization may not be required to achieve probiotic benefits. Therefore, the probiotic potential of the strain AKK PROBIO should not be affected by its weak adherence. We used a classical probiotic colonization evaluation method; however, as *A. muciniphila* does not use glucose as a limiting carbon source, there may be some limitations in the method. 

Antibiotic resistance is a significant concern when assessing potential probiotic candidates [42]. The EFSA has introduced the concept of the qualified presumption of safety, which emphasizes the necessity of thoroughly evaluating the antibiotic resistance genes and horizontal transfer capabilities of biological agents [43]. Thus far, no transferable antibiotic resistance genes for *A. muciniphila* have been documented. We chose seven widely used antibiotics, guided by the predictions of CARD and ResFinder, to comprehensively assess antibiotic resistance. Our results diverged from those reported by Cozzolino et al., as they observed resistance of *A. muciniphila* to chloramphenicol [26]. In our study, *A. muciniphila* PROBIO exhibited sensitivity to most of the common antibiotics, except moxifloxacin. These results underscore the strain-specific nature of antibiotic resistance. The CARD database and ResFinder databases were used to explore the drug resistance of *A. muciniphila* PROBIO across a broader spectrum. The prediction yielded only three resistance genes belonging to two categories, indicating that AKK PROBIO is susceptible to the majority of antibiotics. According to the study by Filardi et al., all strains of *A. muciniphila* harbor the gene *adeF*, encoding for a subunit of the resistance-nodulation-cell division efflux pump system [44].

However, it is important to acknowledge that relying solely on the antibiotic resistance database may lead to erroneous conclusions, necessitating more antibiotic sensitivity tests to assess the safety of AKK PROBIO in subsequent investigations. Based on the antibiotic susceptibility evaluation tests, it was found that AKK PROBIO was indeed resistant to moxifloxacin. All *A. muciniphila* strains, including the type strain, showed low sensitivity to ciprofloxacin [44]. Both moxifloxacin and ciprofloxacin are quinolone antibiotics, so *A. muciniphila* may be innately resistant to quinolone antibiotics. The *adeF* gene of the strain was analyzed by bioinformatics as a possible cause of moxifloxacin resistance, and experimental validation was carried out to explore the non-transferability of resistance genes in *A. muciniphila*. *E. faecalis* is an opportunistic pathogen with a propensity to acquire resistance determinants through horizontal gene transfer [45] and is often used as a receptor for antibiotic transfer experiments [46]. *E. faecalis* ATCC 29212, which is sensitive to moxifloxacin, was used as a receptor strain, and the change in MIC value of this strain was observed by a gradient MH dish with moxifloxacin resistance to determine whether this receptor strain could obtain the moxifloxacin resistance gene from *A. muciniphila*. No change in the MIC value of *E. faecalis* ATCC 29212 against moxifloxacin was observed under different binding environments. The moxifloxacin resistance gene *adeF* of the donor strain AKK PROBIO was not transferable. 

The findings from the in vitro genotoxicity assessments indicated that AKK PROBIO does not exhibit mutagenic properties, as determined by the bacterial reverse mutation test, nor does it demonstrate clastogenic or aneugenic effects, as determined by the in vitro mammalian cell micronucleus test. In both a 7-day acute toxicity experiment and a 90-day sub-chronic toxicity experiment, no adverse effects were observed in relation to the test items in clinical observations: body weight, food intake, water intake, routine blood parameters, organ weight, histopathology, glucose, and clinical chemistry parameters. Therefore, the no-observed-adverse-effect level (NOAEL) for the 90-day study was concluded to be 9.2 × 10^9^ cells/kg body weight/day (the highest dose tested), corresponding to doses of 6.4 × 10^11^ viable bacteria for an average 70 kg human being. The value obtained for viable AKK PROBIO cells was within the range of typical values for probiotics.

Genome-based safety assessments of probiotics are gaining popularity due to their cost-efficiency and speed. Whole-genome sequencing (WGS) of individual probiotic strains provides more information about gene function, such as putative virulence factors and antibiotic resistance genes [47]. In the present study, the gene function prediction results showed that the majority of these genes exhibit multifunctionality, primarily encompassing the synthesis of structural proteins and proteins involved in physiological activities, such as the general metabolic processes of the cell. None of these genes have previously been documented in relation to the pathogenesis of *A. muciniphila*. The virulence factor and pathogenicity were the main concerns when considering *A. muciniphila* for probiotic definition. Only one putative virulence factor gene, *tufA*, was annotated in the genome of AKK PROBIO through comparative analysis with the VFD database. The gene *tufA* encodes the translation elongation factor Tu 1, which primarily participates in protein translation and is widely present in prokaryotes [48]. There has been no report about the role of *tufA* in causing disease. No other strain-specific virulence factors of *A. muciniphila* PROBIO were detected, and no virulence-related factors were found to be associated with the production and secretion systems of toxins.

## 5. Conclusions

In summary, we evaluated the strain *A. muciniphila* PROBIO in terms of the physicochemical characteristics of the cell surface, biofilm formation, resistance to antibiotics, tolerance to simulated gastrointestinal fluids, mutagenicity, acute toxicity, sub-chronic toxicity, and conjugal transfer of moxifloxacin resistance. Overall, *A. muciniphila* PROBIO in its viable form was shown to (i) have a better GI tolerance, suggesting its desirable probiotic properties; (ii) have no mutagenic or clastogenic properties; (iii) have no risk of spreading antibiotic resistance genes; (iv) have no toxicological effect, as confirmed by acute and sub-chronic toxicity studies; (v) have no potential virulence factors or any pathogenic properties; and (vi) its moxifloxacin resistance does not metastasize. Considering these findings together, *A. muciniphila* PROBIO is most likely safe for use as a probiotic candidate. This study supplements the initial safety assessment work already carried out for *A. muciniphila* and contributes to the development of next-generation probiotics.

## Figures and Tables

**Figure 1 foods-13-00442-f001:**
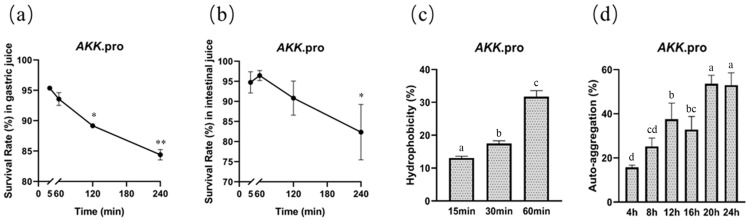
General characteristics of *A. muciniphila* PROBIO. (**a**) Strain tolerance to artificial gastric fluid. (**b**) Strain tolerance to artificial intestinal fluid. (**c**) Adhesion of AKK PROBIO to hydrocarbons. (**d**) Auto-aggregation percentages of AKK PROBIO. The asterisk (*/**) indicates a significant (*p* < 0.05) or highly significant (*p* < 0.01) difference. Different letters indicate significant differences (*p* < 0.05).

**Figure 2 foods-13-00442-f002:**
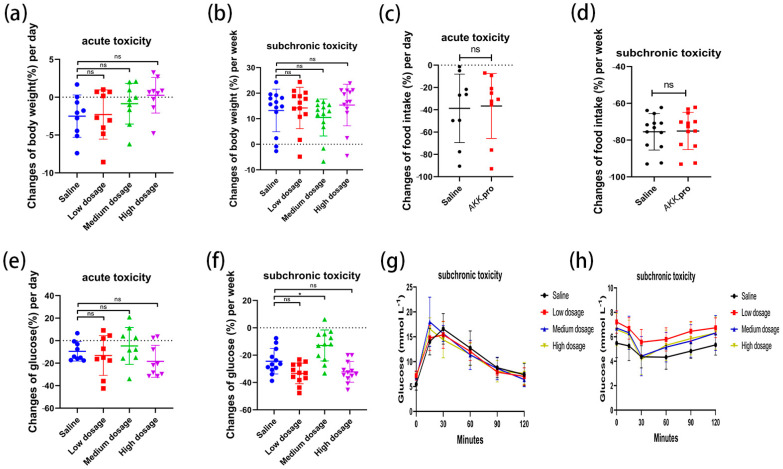
Changes in glucose metabolism in acute and sub-chronic toxicities of *A. muciniphila* PROBIO in normal mice. (**a**,**b**) Body weight; (**c**,**d**) food consumption; (**e**,**f**) blood sugar; (**g**) glucose tolerance; (**h**) insulin tolerance. “ns” indicates no significant difference between the two groups (*p* > 0.05); asterisk (*) indicates *p* < 0.05.

**Figure 3 foods-13-00442-f003:**
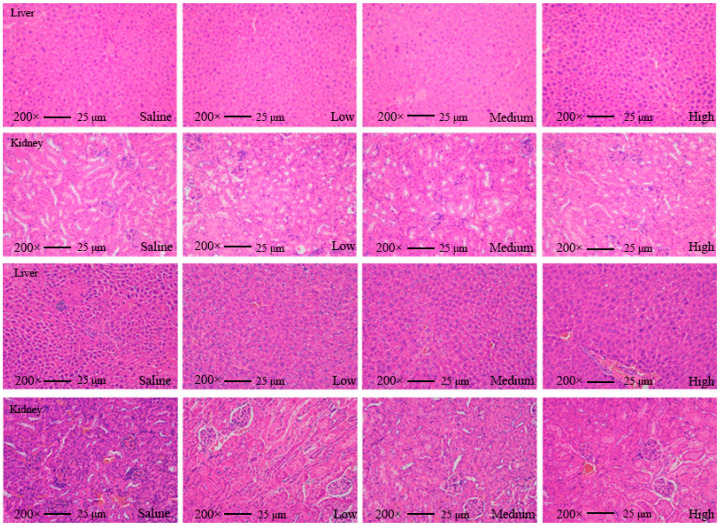
Tissue slice images of the liver and kidney from mice in the acute toxicity experiment (top two) and sub-chronic toxicity experiment (bottom two) at different concentrations of *A. muciniphila* PROBIO.

**Figure 4 foods-13-00442-f004:**
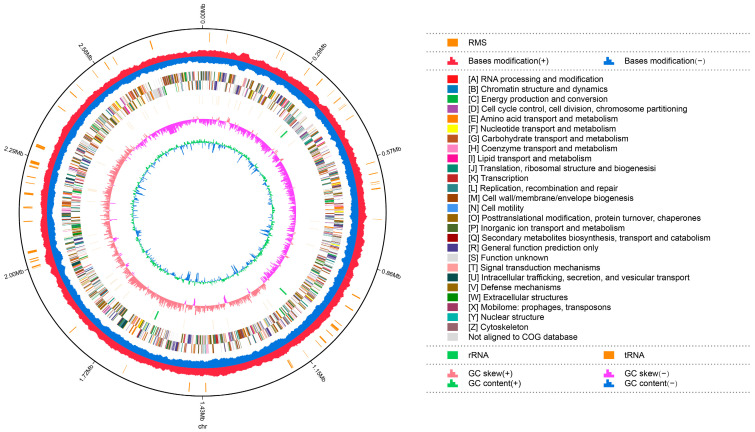
Genome map of *A. muciniphila* PROBIO.

**Figure 5 foods-13-00442-f005:**
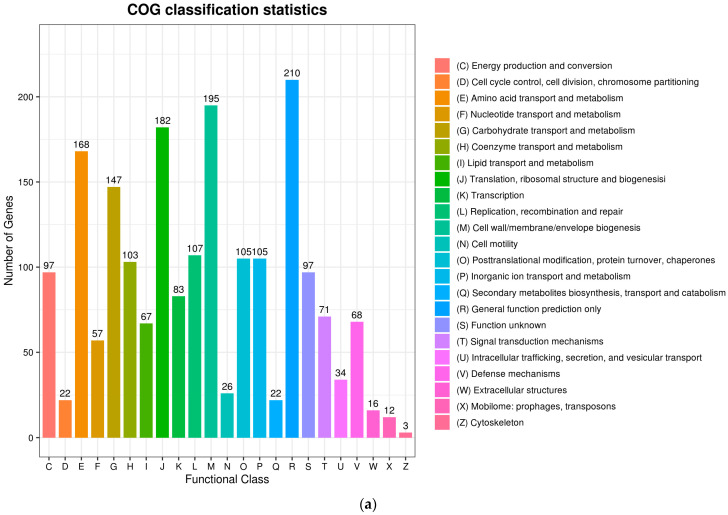
Functional distribution of COG annotation (**a**), KEGG annotation (**b**), and GO annotation (**c**). The X axis represents the number of unigenes. Unigenes were grouped into 23 KOG categories in the COG functional classification.

**Table 1 foods-13-00442-t001:** Summary of reverted colonies of *Salmonella typhimurium* (Ames test) with and without metabolic activation with S9.

Concentrations	*S. typhimurium*								*E. coli*	
	TA97a		TA98		TA100		TA1535		WP2 uvrA	
	−S9	+S9	−S9	+S9	−S9	+S9	−S9	+S9	−S9	+S9
Distilled water	122.0 ± 30.7	126.2 ± 22.1	41.4 ± 8.1	35.0 ± 6.4	144.6 ± 25.5	109.0 ± 43.8	12.0 ± 4.4	16.0 ± 6.5	122.4 ± 19.4	162.0 ± 32.4
AKK PROBIO	98.2 ± 7.7	101.8 ± 7.7	36.8 ± 7.5	31.6 ± 6.8	61.6 ± 12.9	52.0 ± 9.6	10.0 ± 3.9	12.4 ± 2.9	113.6 ± 5.8	130.8 ± 8.4
Positive control 1	1169.4 ± 181.5	-	1525.0 ± 271.7	-	-	-	-	-	-	-
Positive control 2	-	-	-	-	1347.0 ± 202.3	-	-	-	613.4 ± 61.9	-
Positive control 3	-	-	-	-	-	-	141.8 ± 26.6	-	-	-
Positive control 4	-	1288.4 ± 296.3	-	1398.6 ± 309.1	-	1247.4 ± 59.7	-	163.2 ± 9.9	-	548.6 ± 111.8

Data are presented as revertant colonies per plate (mean ± standard deviation). The positive controls 1–4 were: Dixon, methyl methanesulfonate, sodium azide, and 2-aminofluorene, respectively.

**Table 2 foods-13-00442-t002:** In vitro mammalian cell micronucleus test with *A. muciniphila* PROBIO.

Group	PCE (%)	NCE Ratio	PCE Ratio
Negative control	36.34 ± 4.63	0.06 ± 0.09	0.09 ± 0.04
Low dose	36.28 ± 7.83	0.03 ± 0.04	0.11 ± 0.06
Mid dose	34.83 ± 4.73	0.02 ± 0.05	0.12 ± 0.06
High dose	32.73 ± 4.46	0.04 ± 0.03	0.12 ± 0.08
Positive control	24.6 ± 5.90	0.33 ± 0.10 ***	4.76 ± 2.13 ***

PCE, polychromatic erythrocyte; NCE, polychromatic erythrocyte; ***, *p* < 0.001.

**Table 3 foods-13-00442-t003:** Antibiotic resistance of *A. muciniphila* PROBIO to different antibiotics.

**Antibiotics**	**Judging Rule**			**Tested Strain**		**Quality Control Strain**		
				***A. muciniphila* PROBIO**		***B. fragilis* ATCC 25285**		
	**S**	**I**	**R**	**MIC (μg/mL)**	**Result**	**Quality Control Range**	**MIC (μg/mL)**	**Within the Quality Control Range**
Clindamycin	≤2	4	≥8	0.5	Sensitivity	0.5–2	2	Yes
Meropenem	≤4	8	≥16	0.06	Sensitivity	0.03–0.25	0.25	Yes
Tetracycline	≤4	8	≥16	4	Sensitivity	0.125–0.5	0.5	Yes
Moxifloxacin	≤2	4	≥8	>32	Resistance	0.125–0.5	0.5	Yes
Chloramphenicol	≤8	16	≥32	2	Sensitivity	2–8	8	Yes
**Antibiotics**	**Judging Rule**			**Tested Strain**		**Quality Control Strain**		
				***A. muciniphila* PROBIO**		***S. aureus* ATCC 29213**		
	**S**	**I**	**R**	**MIC (μg/mL)**	**Result**	**Quality Control Range**	**MIC (μg/mL)**	**Within the Quality Control Range**
Ceftriaxone	≤16	32	≥64	2	Sensitivity	1–8	4	Yes

S, susceptible; I, intermediate; R, resistance; MIC, minimal inhibitory concentration.

**Table 4 foods-13-00442-t004:** Summary of hematological values in the study of acute and sub-chronic toxicity in mice (*n* = 8).

Groups	Blood Routine Indexes	Saline	Low Dose	Medium Dose	High Dose	Unit
Acute toxicity	white blood cells	0.7 ± 0.2	0.9 ± 0.4	1.1 ± 0.5	0.7 ± 0.3	10^9^/L
	red blood cells	1.3 ± 0.6	2.1 ± 0.6	2.2 ± 0.8	1.4 ± 0.5	10^12^/L
	hemoglobin content	85.8 ± 15.3	87.5 ± 16.1	96.0 ± 14.1	110.0 ± 11.3	g/L
	platelets	1028.8 ± 408.3	921.0 ± 360.8	891.5 ± 316.9	1012.5 ± 344.3	10^9^/L
	lymphocytes	0.4 ± 0.2	0.8 ± 0.5	0.9 ± 0.4	0.5 ± 0.3	10^9^/L
	neutrophils	0 ± 0	0.1 ± 0.07	0.1 ± 0.08	0 ± 0	10^9^/L
	platelets–large cells	314.5 ± 107.4	273.5 ± 95.2	258.5 ± 86.9	228.5 ± 64.3	10^9^/L
	hematocrit value	7.3 ± 2.31	11.7 ± 2.7	11.7 ± 3.6	7.4 ± 2.6	%
	mean corpuscular hemoglobin concentration	1667.5 ± 303.1	802.0 ± 427.0	892.0 ± 422.8	1579.0 ± 438.4	g/L
Sub-chronic toxicity	white blood cells	1.8 ± 0.2	1.5 ± 0.4	1.6 ± 0.5	1.3 ± 0.3	10^9^/L
	red blood cells	7.4 ± 0.5	7.3 ± 0.3	7.6 ± 0.3	7.6 ± 0.5	10^12^/L
	hemoglobin content	107.5 ± 4.3	105.0 ± 3.2	110.6 ± 4.1	107.2 ± 3.4	g/L
	platelets	500.1 ± 62.5	476.3 ± 75.4	475.6 ± 50.6	508.2 ± 71.7	10^9^/L
	lymphocytes	1.7 ± 0.6	1.4 ± 0.4	1.5 ± 0.7	1.1 ± 0.2	10^9^/L
	neutrophils	0.1 ± 0.01	0.1 ± 0.05	0.1 ± 0.03	0.2 ± 0.02	10^9^/L
	platelets–large cells	61.2 ± 30.2	101.8 ± 41.3	98.3 ± 55.3	71.7 ± 43.1	10^9^/L
	hematocrit value	37.9 ± 2.1	36.6 ± 1.2	38.1 ± 0.9	37.6 ± 1.0	%
	mean corpuscular hemoglobin concentration	284.3 ± 12.3	286.5 ± 9.1	290.3 ± 17.5	285.2 ± 13.4	g/L

**Table 5 foods-13-00442-t005:** Biochemical indicators in the serum and liver in acute and sub-chronic toxicity experiments (*n* = 8).

Groups	Indicators	Saline		Low Dose		Medium Dose		High Dose	
		Serum	Liver	Serum	Liver	Serum	Liver	Serum	Liver
Acute toxicity	BG (mmol/L) *	8.5 ± 1.6	9.1 ± 2.1	8.0 ± 1.2	8.6 ± 1.5	8.3 ± 0.7	8.7 ± 1.6	8.3 ± 1.3	9.1 ± 1.1
	TG (mmol/L)	1.5 ± 0.6	0.4 ± 0.05	1.4 ± 0.6	0.3 ± 0.03	1.2 ± 0.3	0.3 ± 0.03	1.4 ± 0.4	0.3 ± 0.03
	TC (mmol/L)	4.9 ± 2.0	0.05 ± 0.03	4.7 ± 1.6	0.04 ± 0.01	5.2 ± 1.7	0.04 ± 0.01	4.2 ± 0.8	0.03 ± 0.01
	BA (μmol/L)	3.3 ± 0.7	56.2 ± 35.6	3.3 ± 0.6	49.8 ± 35.3	2.9 ± 0.4	45.7 ± 25.7	2.8 ± 0.3	52.7 ± 35.2
	TP (μg/mL)	241.4 ± 28.5	781.9 ± 71.0	260.1 ± 80.3	806.2 ± 47.9	217.6 ± 59.0	812.1 ± 37.3	240.9 ± 70.8	813.1 ± 69.4
	ALT (U/L)	15.3 ± 5.5	9.2 ± 2.7	17.3 ± 4.8	9.9 ± 2.6	15.8 ± 3.8	9.1 ± 3.0	18.0 ± 4.9	8.9 ± 3.4
	AST (U/L)	40.7 ± 9.0	10.1 ± 1.9	42.5 ± 3.8	9.6 ± 2.0	43.5 ± 8.7	10.1 ± 2.2	48.3 ± 7.5	9.3 ± 2.1
	Cr (μmol/L)	15.7 ± 4.9	-	20.7 ± 8.1	-	14.2 ± 4.2	-	15.8 ± 5.0	-
	BUN (mmol/L)	5.7 ± 1.8	-	5.3 ± 1.4	-	5.7 ± 2.0	-	5.7 ± 1.7	-
Sub-chronic toxicity	BG (mmol/L)	8.4 ± 1.5	11.8 ± 2.3	7.9 ± 1.3	11.2 ± 2.1	8.5 ± 1.6	13.0 ± 1.4	8.3 ± 1.2	11.5 ± 2.0
	TG (mmol/L)	1.2 ± 0.4	1.7 ± 0.6	1.1 ± 0.2	2.6 ± 1.2	1.3 ± 0.5	2.4 ± 1.0	1.8 ± 0.9	1.9 ± 0.7
	TC (mmol/L)	1.8 ± 0.7	0.3 ± 0.1	1.6 ± 0.7	0.2 ± 0.1	1.8 ± 0.8	0.2 ± 0.1	2.2 ± 0.5	0.2 ± 0.1
	BA (μmol/L)	5.0 ± 1.8	2.6 ± 0.7	5.7 ± 1.2	2.1 ± 0.5	4.1 ± 1.0	2.4 ± 0.7	5.8 ± 2.0	2.8 ± 0.5
	TP (μg/mL)	1629 ± 271.1	595.0 ± 123.1	1512 ± 261.7	621.5 ± 90.2	1684 ± 225.1	654.4 ± 86.9	1516 ± 209.9	628.2 ± 89.2
	ALT (U/L)	10.8 ± 4.1	9.2 ± 2.7	10.7 ± 3.8	9.9 ± 2.6	11.1 ± 4.1	9.1 ± 2.8	10.8 ± 4.0	9.0 ± 2.3
	AST (U/L)	44.3 ± 10.1	268.6 ± 29.5	52.6 ± 8.7	240.9 ± 50.4	48.7 ± 12.5	232.2 ± 25.1	47.2 ± 7.5	280.4 ± 27.5
	Cr (μmol/L)	13.9 ± 2.9	-	14.1 ± 3.1	-	13.9 ± 1.6	-	13.9 ± 2.1	-
	BUN (mmol/L)	3.9 ± 0.7	-	4.5 ± 1.6	-	4.3 ± 1.6	-	4.0 ± 0.8	-

Abbreviations: BG, blood glucose; TG, triglycerides; TC, total cholesterol; BA, bile acid; TP, total protein; ALT, alanine aminotransferase; AST, aspartate aminotransferase; Cr, creatinine; BUN, blood urea nitrogen; *, Units: BG, TG, and TC (mmol/L serum, or mmol/gprot liver); -, untested.

**Table 6 foods-13-00442-t006:** Summary of relative organ weights (g/kg bw) in acute and sub-chronic toxicity experiments (*n* = 8).

		Heart	Liver	Spleen	Kidney	Thymus	Brain	Testicle	Lung	Stomach	Intestines
Acute toxicity	Saline	5.8 ± 1.0	43.7 ± 7.5	2.4 ± 0.5	13.6 ± 2.5	2.7 ± 0.5	9.9 ± 1.6	6.2 ± 0.5	6.1 ± 0.8	17.3 ± 5.6	101.4 ± 11.3
	Low dosage	5.6 ± 0.8	44.3 ± 6.6	3.1 ± 0.5	14.2 ± 2.6	2.5 ± 0.3	10.7 ± 2.1	6.5 ± 0.4	6.3 ± 0.9	16.5 ± 7.3	97.7 ± 7.5
	Medium dosage	4.8 ± 1.6	47.4 ± 7.7	2.5 ± 0.2	13.1 ± 1.9	3.6 ± 0.6	11.9 ± 2.2	6.1 ± 0.8	5.9 ± 0.6	15.9 ± 5.1	98.5 ± 6.5
	High dosage	4.7 ± 1.7	48.0 ± 3.9	2.8 ± 0.8	12.4 ± 1.3	3.5 ± 0.8	10.6 ± 4.8	7.2 ± 0.7	6.7 ± 1.1	15.8 ± 6.3	113.3 ± 9.3
Subchronic toxicity	Saline	4.8 ± 0.5	38.7 ± 4.7	2.1 ± 0.6	11.6 ± 2.0	2.6 ± 0.9	9.1 ± 3.3	6.0 ± 0.9	6.2 ± 0.9	16.1 ± 6.6	80.3 ± 12.7
	Low dosage	4.7 ± 0.7	37.8 ± 2.3	2.5 ± 0.8	12.6 ± 1.9	2.6 ± 0.9	10.2 ± 2.1	6.2 ± 1.3	5.9 ± 0.6	15.8 ± 7.2	79.6 ± 16.0
	Medium dosage	4.7 ± 0.5	36.2 ± 4.3	2.6 ± 0.5	12.2 ± 1.8	2.4 ± 0.2	9.0 ± 2.0	6.4 ± 0.7	5.2 ± 0.8	14.4 ± 4.3	80.2 ± 10.8
	High dosage	5.0 ± 0.3	38.3 ± 6.4	2.8 ± 0.7	12.0 ± 1.2	3.6 ± 1.4	9.2 ± 1.7	5.8 ± 0.6	5.6 ± 0.4	15.7 ± 2.5	87.6 ± 8.5

**Table 7 foods-13-00442-t007:** Transfer of the moxifloxacin resistance gene.

Growth in Moxifloxacin Medium (μg/mL)		0	0.05	0.125	0.5	2	4	8
1:1 Liquid	*A. muciniphila* and *E. faecalis*	+	+	+	−	−	−	−
	*E. faecalis* alone	+	+	+	−	−	−	−
10:1 Liquid	*A. muciniphila* and *E. faecalis*	+	+	+	−	−	−	−
	*E. faecalis* alone	+	+	+	−	−	−	−
1:1 Agar	*A. muciniphila* and *E. faecalis*	+	+	+	−	−	−	−
	*E. faecalis* alone	+	+	+	−	−	−	−
10:1 Agar	*A. muciniphila* and *E. faecalis*	+	+	+	−	−	−	−
	*E. faecalis* alone	+	+	+	−	−	−	−
1:1 0.22 μm Filter	*A. muciniphila* and *E. faecalis*	+	+	+	−	−	−	−
	*E. faecalis* alone	+	+	+	−	−	−	−
10:1 0.22 μm Filter	*A. muciniphila* and *E. faecalis*	+	+	+	−	−	−	−
	*E. faecalis* alone	+	+	+	−	−	−	−

## Data Availability

The data used in the current study are available upon reasonable request to the corresponding author. The data are not publicly available due to privacy restrictions.

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
