# Peer review of "Characterization and Preliminary Safety Evaluation of Akkermansia muciniphila PROBIO"

_foods, 2024, doi:10.3390/foods13030442_

Round 1

Reviewer 1 Report

Comments and Suggestions for Authors

After reviewing the manuscript titled “Characterization and Preliminary Safety Evaluation of Akkermansia muciniphila PROBIO” I have the following observations and comments:

The manuscript is well written, comprehensive, structurally and scientifically sound with proper flow, however, there are some aspects of the manuscript that needs more attention and additional information should be provided:

1.       If you are writing about a new strain, in this case its Akkermansia muciniphila PROBIO, information about this strain should be deposited in the public database (for example NCBI) and accession number should be provided in the manuscript.

2.       Fig. 1  - statistical significance should be presented in the figures.

Apart from this, the manuscript is of scientific interest and these points need be adjusted.

Author Response

Response to Reviewer

Reviewer:

Comments for the Author:

The manuscript is well written, comprehensive, structurally and scientifically sound with proper flow, however, there are some aspects of the manuscript that needs more attention and additional information should be provided

Response to reviewer:

We very much appreciate your valuable suggestions and advice on our submitted manuscript. We found your comments to be very helpful and constructive, and revisions have been made accordingly. All changes made to the text are highlighted in red so that you can easily identify them. With regard to your comments and suggestions, we wish to respond as follows:

Comments:

Comment 1: If you are writing about a new strain, in this case its Akkermansia muciniphila PROBIO, information about this strain should be deposited in the public database (for example NCBI) and accession number should be provided in the manuscript.

Response: Thank you for your valuable suggestion. The complete genome sequences of Akkermansia muciniphila PROBIO are available from NCBI SRA accession number PRJNA1065580.

Comment 2: Fig. 1  - statistical significance should be presented in the figures.

Response: Thanks for your good recommendation. We have added the statistical significance in Figure 1 (as shown in the attachment) and re-uploaded the picture.

We hope that these responses will now meet with your approval.

Once again, thank you very much for your time and energy on this manuscript.

Reviewer 2 Report

Comments and Suggestions for Authors

The manuscript deals with the evaluation on a newly isolated A. muciniphila strain from human feces in terms of its safety, physiological and biochemical characteristics. The topic is interesting, especially since limited information is available for A. muciniphila which is considered as a next-generation probiotic organism. The manuscript is overall well-written, however certain parts require improvements. Specific comments are listed below:

- In section 2.2. Viable bacterial count was measured at 0 and 3h. In the equation 1 and 3 hours are mentioned instead. Needs clarification.

- The centrifugal conditions need to be expressed in g and not rpm. 

- A reference is needed for the categorisation of biofilm formation ability based on OD measurements. 

- Please correct to Salmonella Typhimurium instead of Salmonella typhimurium

- How were the antibiotics selected for testing in 2.6.? Why these specific antibiotics and not others, commonly used?

- Line 294: After 240min means 4h in total. In the materials and methods 3h were mentioned instead.

- Line 299-300: You have used an in vitro method to indirectly assess colonisation. Wouldn't it be better to use cell-lines instead? Statement in lines 299-300 needs to be rephrased because its inaccurate.

- Table e of Figure 1 can be completely eliminated since it does not give any value. Results can be mentioned just in the text.

- Line 450-451: Potential probiotic properties, why data not shown?

Which are the probiotic properties you are referring to? Is the survival to gastric and intestinal juice, auto-aggregation, hydrophobicity part of these properties? 

Comments on the Quality of English Language

In some parts of the document, 3rd person and passive voice needs to be used.

e.g. "It was isolated" instead of "we isolated".

Minor editing is required overall. 

Author Response

Response to reviewer:

We very much appreciate your valuable suggestions and advice on our submitted manuscript. We found your comments to be very helpful and constructive, and revisions have been made accordingly. All changes made to the text are highlighted in red so that you can easily identify them. With regard to your comments and suggestions, we wish to respond as follows:

Comments:

Comment 1: In section 2.2. Viable bacterial count was measured at 0 and 3h. In the equation 1 and 3 hours are mentioned instead. Needs clarification

Response: We are very sorry we didn't make it clear. According to your suggestion, we have made changes in the revised manuscript. (Line 120-124).

Comment 2: The centrifugal conditions need to be expressed in g and not rpm.

Response: The centrifugal conditions have been checked and corrected.

Comment 3: Please correct to Salmonella Typhimurium instead of Salmonella typhimurium.

Response: Thank you for your nice suggestion. The typo is revised in our revised manuscript.

Comment 4: How were the antibiotics selected for testing in 2.6.? Why these specific antibiotics and not others, commonly used?

Response: Clinical and Laboratory Standards Institute (CLSI) is the gold standard for antibiotic susceptibility test and the Akkermansia muciniphila are gram-negative anaerobes. According to the CLSI guidelines, the test that need to be carried out for gram-negative anaerobes are as follows: lincosamides (clindamycin), penicillins (meropenem), tetracyclines (tetracycline), fluoroquinolones (moxifloxacin), phenicols (chloramphenicol), and cephem (ceftriaxone).

Ref.:

  1. Performance Standards for Antimicrobial Susceptibility Testing, 33rd ed. CLSI supplement M100. Clinical and Laboratory Standards Institute; 2014;
  2. Methods for Antimicrobial Susceptibility Testing of Anaerobic Bacteria 9th ed. CLSI standard M11.Wayne, PA: Clinical and Laboratory Standards Institute; 2007.

Comment 5: Line 294: After 240min means 4h in total. In the materials and methods 3h were mentioned instead.

Response: We are sorry that our expression is not clear, which caused your misunderstanding. We have made changes in the revised manuscript.

Comment 6: Line 299-300: You have used an in vitro method to indirectly assess colonisation. Wouldn't it be better to use cell-lines instead? Statement in lines 299-300 needs to be rephrased because its inaccurate

Response: Thank you for your nice suggestion. The microtiter-plate test are the most frequently used techniques for quantifying biofilm formation. The method used in this manuscript is an improved biofilm measurement method based on the microtiter-plate test. Compared with cell experiments, it can detect the biofilm formation ability of strains simply, quickly and effectively. Moreover, we further described the corresponding statement accurately in the revised manuscript. (Line 298-300).

Ref.:

  1. Stepanović S ,Vuković D ,Dakić I , et al.A modified microtiter-plate test for quantification of staphylococcal biofilm formation[J].Journal of Microbiological Methods,2000,40(2):175-179.
  2. Cozzolino A ,Vergalito F ,Tremonte P , et al.Preliminary Evaluation of the Safety and Probiotic Potential of Akkermansia muciniphila DSM 22959 in Comparison with Lactobacillus rhamnosus GG[J].Microorganisms,2020,8(2):189.

Comment 7: Table e of Figure 1 can be completely eliminated since it does not give any value. Results can be mentioned just in the text.

Response: Thanks for your good recommendation. We have deleted the Table e of Figure 1 in Figure 1 (as shown below) and re-uploaded the picture.

Comment 8: Line 450-451: Potential probiotic properties, why data not shown?

Which are the probiotic properties you are referring to? Is the survival to gastric and intestinal juice, auto-aggregation, hydrophobicity part of these properties?

Response: For Akkermansia muciniphila PROBIO we explored the mechanism of its probiotic effect against colorectal cancer (paper accepted), ulcerative colitis (paper under review), this study mainly focuses on the safety evaluation of AKK PROBIO so it does not refer to its more probiotic efficacy research data.

Comment 9: In some parts of the document, 3rd person and passive voice needs to be used.

Response: Thank you for your valuable suggestion. According to your suggestion, we revised in the manuscript.

We hope that these responses will now meet with your approval.

Once again, thank you very much for your time and energy on this manuscript.

Reviewer 3 Report

Comments and Suggestions for Authors

Review: „Characterization and Preliminary Safety Evaluation of Akkermansia muciniphilaPROBIO”

The authors describe in their manuscript a fundamental evaluation of the safety of the novel bacteria strain Akkermansia muciniphila PROBIO. Therefore, the authors determined basic features such as strain tolerance against gastric and artificial intestinal fluids, the adhesion to hydrocarbons, resistances against antibiotics, or the auto aggregation to each other. Subsequently, a reverse mutation test has been carried out. All experiments have been underpinned by the genetic characteristics of the strain. Additionally, the toxicity has been tested in a mouse model. Based on test guidelines, the latter findings have been underpinned by a sub-chronic oral toxicity test.

The manuscript is very well written and it has no serious flaws. Since I´m not a mother tongue, I feel unsure about judging the spelling and English of the authors. In my opinion, the manuscript only needs minor spelling and grammar changes. 

General remarks:

Line 124/equation (1): In the presence of gastric and intestinal fluids, the fraction highlighted in equation 1 are the viable counts of the bacteria after 3h of cultivation in the numerator and the tvc of the same bacteria after 1h cultivation under the same conditions in the denominator. The text introducing that process step talks about a tvc after 0h. Please correct and introduce the right value.

Line 185: What are the prevailing regulatory guidelines you are talking about? Do we need a reference to follow? Are the guidelines under ref. [32] the same?

Figure 2: Please enlarge. The captions are hard to read.

Based on the conscientious review, I think the manuscript needs minor revisions. It is good work.

Comments on the Quality of English Language

The manuscript is very well written and it has no serious flaws. Since I´m not a mother tongue, I feel unsure about judging the spelling and English of the authors. In my opinion, the manuscript only needs minor spelling and grammar changes.

Author Response

Response to reviewer:

Thank you for these comments, we appreciate you taking the time to review the manuscript. The revisions have been made accordingly. All changes made to the text are highlighted in red so that you can easily identify them. With regard to your comments and suggestions, we wish to respond as follows:

Comment 1: Line 124/equation (1): In the presence of gastric and intestinal fluids, the fraction highlighted in equation 1 are the viable counts of the bacteria after 3h of cultivation in the numerator and the tvc of the same bacteria after 1h cultivation under the same conditions in the denominator. The text introducing that process step talks about a tvc after 0h. Please correct and introduce the right value.

Response: Thank you very much for your careful review. In the revised manuscript, we checked and revised the corresponding content.

Comment 2: Line 185: What are the prevailing regulatory guidelines you are talking about? Do we need a reference to follow? Are the guidelines under ref. [32] the same?

Response: We are sorry that our expression is not clear, which caused your misunderstanding. The current regulatory standards written in this paper are consistent with the guidelines under ref. [32].

Comment 3: Figure 2: Please enlarge. The captions are hard to read.

Response: Thanks for your valuable suggestion. We have modified the captions in the Figure 2 to enable the reader to clearer, and re-uploaded the picture.

Once again, thank you very much for your attention to this manuscript.

We are more than willing to answer any of your questions concerning this paper and our research.